# High-Field Magnetic Resonance Imaging of the Temporomandibular Joint Low Agreement with Clinical Diagnosis in Asymptomatic Females

**DOI:** 10.3390/diagnostics13121986

**Published:** 2023-06-06

**Authors:** Milica Jeremic Knezevic, Aleksandar Knezevic, Jasmina Boban, Aleksandra Maletin, Bojana Milekic, Daniela Djurovic Koprivica, Tatjana Puskar, Robert Semnic

**Affiliations:** 1Faculty of Medicine Novi Sad, University of Novi Sad, 21000 Novi Sad, Serbia; aleksandar.knezevic@mf.uns.ac.rs (A.K.); jasmina.boban@mf.uns.ac.rs (J.B.); aleksandra.maletin@mf.uns.ac.rs (A.M.); bojana.milekic@mf.uns.ac.rs (B.M.); daniela.djurovic-koprivica@mf.uns.ac.rs (D.D.K.); tatjana.puskar@mf.uns.ac.rs (T.P.); 2Medical Rehabilitation Clinic Clinical Centre of Vojvodina, 21000 Novi Sad, Serbia; 3Institute for Oncology, Center for Imaging Diagnostic, 21208 Sremska Kamenica, Serbia; 4Dentistry Clinic of Vojvodina, 21000 Novi Sad, Serbia; 5Department of Radiology, Upssala University Hospital, 752 36 Upssala, Sweden; robert.semnic@akademiska.se

**Keywords:** temporomandibular joint, magnetic resonance imaging, high-field MRI, agreement, RDC/TMD, temporomandibular disorders

## Abstract

(1) Background: The aim of this study was to investigate the agreement between a clinical diagnosis based on research diagnostic criteria/temporomandibular disorders (RDC/TMD) and high-field magnetic resonance imaging (MRI) findings of temporomandibular joints (TMJs) in asymptomatic females. (2) Methods: A prospective study on 100 females (200 TMJs) was performed, using clinical examinations (RDC/TMD) and same-day MRIs of TMJs on a 3T MR unit. The inclusion criteria were as follows: females, age > 18, the presence of upper and lower incisors, and an understanding of the Serbian language. Descriptive statistics (means and standard deviations) and ANOVA with a post hoc Tukey test for differences among the patient subgroups was performed. The agreement between the clinical and MRI findings was determined using Cohen’s kappa coefficient (k < 0.21 slight, 0.21–0.4 fair, 0.41–0.6 moderate, 0.61–0.8 substantial, and 0.81–1 almost perfect). The statistical significance was set at *p* ≤ 0.05. (3) Results: Normal findings were observed in 86.7%, disc dislocation (DD) was observed in 9.2%, and arthralgia/osteoarthritis/osteoarthrosis was observed in 2.6% of TMJs using RDC/TMD. On the MRI, normal findings were observed in 50.5%, disc dislocation was observed in 16.3%, and arthralgia/osteoarthritis/osteoarthrosis was observed in 23.5% of TMJs. The anterior DD with reduction showed fair agreement of the clinical and MRI findings (k = 0.240, *p* < 0.001) compared with the DD without reduction (k = 0.355, *p* < 0.001). Both showed high specificity (94.9% and 99.4%) but low sensitivity (24.2% and 25.0%). The sensitivity in osteoarthritic changes was low (4.8%), but the specificity remained high (96.2%). (4) Conclusions: The sensitivity of the clinical examination remains low compared with 3T MRI, especially in osteoarthritic changes and anterior DD with reduction. However, the number of false positive diagnoses using RDC/TMD is low in asymptomatic patients. RDC/TMD remains a sensible method for establishing a clinical diagnosis and avoiding the overtreatment of asymptomatic patients.

## 1. Introduction

The term temporomandibular dysfunction (TMD) encompasses several pathological conditions affecting the temporomandibular joint (TMJ), masticatory muscles, and surrounding structures [1]. It is estimated that 50–75% of the general population will present with one TMD sign and 20–25% with one TMD symptom during their lives [2]. Females are almost twice as prone to TMD, and the peak occurrence of these disorders is between 20 and 40 years of age, with pain in the TMJ as the most common symptom [3]. Some authors estimate that the prevalence of TMD is 5–12% in the general population, thus representing the second most frequent location of musculoskeletal pain (after low back pain) and a very frequent cause of absenteeism [4].

The TMJ is a morphologically and functionally complex joint [5,6]. Musculoskeletal disorders of the masticatory system (the TMJ together with masticatory muscles) are hidden behind the name temporomandibular dysfunction and are heterogeneous in etiology and clinical presentation [1]. The TMJ as a ginglymoarthrodial joint distinguishes itself from other synovial joints by several features: articulating surfaces are covered by fibrous cartilage, a dense fibrocartilaginous articular disc is interposed between the condyle and articular eminence, it provides a wide range of motions in three planes, it is nearest to the middle cranial fossa, and it plays a crucial role in mastication, swallowing, respiration, and speaking. The articular disc divides the TMJ into upper and lower joint spaces, which do not communicate in physiological conditions. The TMJ allows rotation in the lower joint space and translation in the upper joint space [7].

Active parts of the orofacial system are both muscles and nerves. The masticatory muscles belong to the cervical muscular chain, and every disturbance in this segment will lead to the reorganization and adaptation of other muscular segments [8]. It is important to emphasize the relationship between poor body posture and the incidence of TMD. However, the causality must not be taken for granted, because it is commonly not possible to establish whether TMD is a cause or a result of body posture deviations. Saito et al. concluded that patients with the most common TMD anterior disc displacement had associated changes in body posture, particularly in the pelvic position, lumbar and thoracic spines, head, and mandibles, in support of the theory that a deviation in one joint subunit may lead to compensations in other joints [9]. Patients with TMD present with an overload in cervical muscles due to increased activity of the masticatory muscles to compensate for the joint disorder. Such an overload can produce mandibular and spinal deviations and cervical hyperlordosis, due to shoulder elevation and head protrusion in patients with TMD. Therefore, the main complaint of TMD—the pain and mandibular opening limitation—may be accompanied by muscle fatigue and postural problems in the patient [10].

The higher prevalence in the female population was thought to be the consequence of different pain tolerance and weaker joint structure, as well as hormonal influences [11]. The prevalence of TMD decreases with age, especially after 55 [12]. Manfredini et al. reported the presence of two peaks of occurrence, one between 30 and 35 and other between 50 and 55 years of age [13], which is in accordance with our study. The incidence of TMD in the general population varies between 18 and 35% [5,14]. The signs and symptoms of TMD include pain in the masticatory region, TMJ, and/or temporal region, limited jaw opening, and sounds (clicks and crepitations).

Disorder of orofacial system function can be a consequence of sudden trauma or a result of destructive forces from the surrounding muscles, instability of the TMJ, joint dysfunctions, psychological disabilities, and non-inflammatory myalgias [15,16].

Magnetic resonance imaging (MRI) is an established standard modality for the assessment of the position and morphology of the disc, as well as the mobility of the condyle and the evaluation of the surrounding structures [17,18]. Furthermore, the 3 Tesla magnetic resonance (MR) scanner is considered superior compared with lower fields using the same examination time for obtaining better quality tomograms [19,20].

Although clinical examination and research diagnostic criteria of temporomandibular disorders (RDC/TMD) are commonly used in clinical practice, an MRI of the TMJ represents the gold standard for disc position and the status of the retrodiscal space [21]. The reliability of the clinical diagnosis remains unclear since a high number of false negative diagnoses are established based on these criteria only.

The aim of this study was to investigate the correlations between the clinical diagnosis based on RDC/TMD and the MRI findings of TMJ in randomly selected, asymptomatic women.

## 2. Materials and Methods

### 2.1. Study Population

An institutional-board-approved prospective study on a total of 100 females (200 TMJs) in a two-year period (January 2018 to January 2020) was performed. Randomly selected patients, who were referred for MR examination of the brain by general physicians due to mild neurological symptoms, were recruited for this study. No patients were referred for an MRI due to symptoms connected to TMD. All patients underwent a clinical examination followed by a same-day MRI on a 3T unit (Siemens Trio Tim, Erlangen, Germany), with a standard MR protocol for the brain and dedicated imaging of TMJ. All participants signed informed consent for entering the study.

The inclusion criteria were as follows: female patients over 18, the presence of upper and lower (to be able to hold the applicator in the mouth) incisors, and a comprehension of the Serbian language (necessary for filling in the forms).

The exclusion criteria were as follows: pregnancy, the presence of a systemic disease involving the TMJ and masticatory muscles (e.g., rheumatoid arthritis, systemic fibrosis etc.), brain tumors, multiple sclerosis, metastatic disease, infectious and inflammatory central nervous system diseases, deprived consciousness, psychiatric disorders, recent facial trauma, contraindications for MRI, and technically inadequate MRI tomograms.

Two patients (four TMJs) were excluded from the study due to technically inadequate MR tomograms.

### 2.2. Clinical Examination

Prior to the MRI, all subjects were examined according to Research Diagnostic Criteria/Temporomandibular Disorders—RDC/TMD. Based on the clinical findings, we divided the participants into the following groups:

Group I:0—normalIIa—dislocation of the disc with reduction (DDWR)IIb—dislocation of the disc without reduction (DDWOR) with limited jaw openingIIc—DDWOR and without limited jaw opening

Group II:0—normalIIIa—arthralgiaIIIb—osteoarthritisIIIc—osteoarthrosis

### 2.3. MRI Protocol

An MRI was performed after the clinical examination, using a TMJ surface coil, with parasagittal and coronal tomograms of both condyles. The proton density sequence was used (time of repetition 1850 ms, time of echo 15 ms, field of view 13 cm, matrix size 256 × 256, slice thickness 2 mm, total time 4:09 min). The TMJ was analyzed in the position of a closed, half-opened, and fully opened jaw. A plastic mechanical mouth opener, made of polymethyl methacrylate in the form of an arrow, with incisures on the upper and lower surfaces (places to put incisors inside) at distances of 15 mm, 20 mm, 25 mm, 30 mm, and 35 mm was used for dynamic imaging of TMJ (Figure 1). These distances represent the possible variations in jaw opening. While inside the MR unit, after having completed sequences performed with a closed jaw, the patient herself placed the applicator in the mouth, according to the instructions given by the radiographer. Sequences with an opened jaw were performed subsequently.

The MRI tomograms were analyzed on digital workstations by two experienced radiologists who were completely unaware of the clinical findings.

### 2.4. Analysis of TMJ

The analysis was based on the detection the presence of the disc, determining its position, shape, and length in positions of closed and fully opened jaws. The position of the disc was classified as normal (disc on 12 o’clock) and pathological (anterior and posterior dislocation) (Figure 2 and Figure 3). Anterior dislocation was further divided into degrees according to the angle between the posterior zone of the disc and the vertical line that goes through the condyle (0–10°, normal; 11–30°, mild; 31–50°, mild moderate; 51–80°, moderate; over 80°, severe).

The shape of the disc was classified as normal (biconcave), biconvex, biplanar, hemiconvex, and deformed (Figure 4).

The length of the disc was measured in millimeters (mm), from the anterior to the posterior border on the sagittal tomogram. Additionally, we measured the angle of the posterior aspect of the articular eminence, using the Frankfort horizontal (FH) plane as a reference point. In the closed jaw position, the highest point of the glenoid fossa (GF) on temporal bone was marked. Drawing the line downward to the most prominent part of the condyle and drawing a line parallel to the FH marked the point on the posterior side of the articular eminence. An angle between the FH and the tangent is formed by drawing a tangential line to that point (Figure 5).

Further, the width and depth of the GF in the sagittal plane were measured. The width of the GF represents the distance between the postglenoid process of the temporal bone to the peak of the articular eminence (in millimeters). The depth of the GF represents the distance between the highest and lowest points of the fossa (in millimeters) (Figure 6).

The mobility of the condyle in the position of the fully opened mouth was observed, and the findings were classified as normal, hypomobility, and hypermobility of TMJ. The algorithm for determining the mobility was as follows: drawing a horizontal tangential line through the peak of the GF and perpendicular to it and a vertical line through the peak of the articular eminence (point 0). The peak of GF was 0, while the peak of the articular eminence was 90. If the condyle was located in the region 0–90° with the opened jaw, it was classified as hypomobile. If it was in the region over 120°, it was classified as hypermobile, and in between those values, the mobility was classified as normal.

### 2.5. Statistical Analysis

The statistical analysis was carried out using SPSS ver. 21.0 (Statistical Package for the Social Sciences, Chicago, IL, USA). The means and standard deviations of the variables were determined with descriptive statistical analyses. The tested variables showed normal distributions: angle of articular eminence (skewness = −0.523, kurtosis = −0.343), disc length (skewness = −0.088, kurtosis = 0.153), and depth of glenoid fossa (skewness = 0.082; kurtosis = −0.729). Differences among the patient subgroups were tested with ANOVA with post hoc analysis using Tukey’s HSD test. Agreement between the clinical and imaging findings for the categorical parameters was determined using Cohen’s kappa coefficient and classified according to Kraemer et al. as follows: lower than 0.21, slight; 0.21–0.4, fair; 0.41–0.6, moderate; 0.61–0.8, substantial; and 0.81–1, almost perfect agreement [22].

The sample size was calculated by GPower (G*Power, Version 3.1.9.6, Heinrich Heine Universität, Düsseldorf, Germany) [23]. The number of participants needed to achieve a statistical power of 0.80 for medium-size effect and an α error probability of 0.05 was 145.

The statistical significance was set at *p* ≤ 0.05.

## 3. Results

### 3.1. Subjects’ Features

Ninety-eight randomly selected female participants, average age 48.64 ± 12.23 years (range 19–65), were included in this study.

### 3.2. Clinical and MRI Findings

Clinical and MRI findings of the TMJs observed are summarized in Table 1.

The majority of TMJs presented with a normal disc shape (136, 69.4%). A biconvex disc was found in seven (3.6%), a biplanar disc was found in 38 (19.4%), a hemiconvex disc was found in nine (4.6%), while a deformed disc was observed in six (3.1%) TMJs.

The longest discs were biplanar and deformed (11.03 ± 2.10 mm and 9.83 ± 3.31 mm, respectively), while the shortest was biconcave (9.60 ± 1.41 mm). 

The findings on the mobility of TMJ are shown in Figure 7. 

The data on the depth of the GF and the angle of articular eminence with regard to disc location are summarized in Table 2. Our study failed to present a significant correlation between the anatomical features of the TMJ and the disc location. Even though the deepest GF was observed in patients with anterior disc dislocation without reduction, there was no significant difference in the depth of the GF among separate groups of patients (F = 1417, *p* = 0.245). Additionally, the steepest articular eminence was observed in patients with anterior DDWOR, but there was no significant difference in the angle of articular eminence between separate groups of patients (F = 0.531, *p* = 0.589). Finally, the longest disc was observed in the group with DDWOR, but these differences in disc length did not reach statistical significance (Table 2).

### 3.3. Agreement Analysis

Anterior DDWR showed lower intermethod agreement (k = 0.240, *p* < 0.001) compared with anterior DDWOR (k = 0.355, *p* < 0.001). Both types showed high specificity (94.9% and 99.4%) but low sensitivity (24.2% and 25.0%). Regarding ostheoarthritic changes, the sensitivity was extremely low (only 4.8%), but the specificity remained very high (Table 3).

## 4. Discussion

Disc dislocation in the general population is found in 8.9–15%, concordantly with our findings (16%). According to RDC/TMD, in symptomatic patients, the incidence of disc dislocation is around 52%, and the same is true for arthralgia/osteoarthritis/osteoarthrosis. In our study, disc dislocation was confirmed in 10.7% and arthralgia/osteoarthritis/osteoarthrosis in 4% of subjects, concordantly with the values reported for the general population incidence [24]. These results were expected, according to the design of the study, with randomly selected, asymptomatic patients recruited.

A common clinical finding of TMJ hypermobility is sound (click) in the joint and pain in the masticatory muscles [25,26]. Hypermobility of TMJ can destabilize the congruence between the disc and the condyle and consequently overload the lateral pterygoid muscle (LPM). Hypomobility of TMJ is a consequence of progressive TMD and often is related to DDWOR [27,28]. Pathophysiologically, hypermobility of the joint is a consequence of the connective tissue weakness, which leads to muscle hypertrophy. Thus, LPM replaces the disc in the role of a chronically overloaded structure. Hypomobility of the TMJ, DDWOR, and osteoarthritis are all present in more severe, symptomatic cases of TMD, often associated with DDWOR, according to the study by Campos et al. [29]. In our study, most joints were of normal mobility (around 70%), with no significant difference in the portions of hypermobile and hypomobile joints (Figure 7). Sonnesen et al. reported a similar incidence of a normal range of mandibular motion but a somewhat higher incidence of hypomobility, presumably due to the different patient selection [30].

There are not many studies on disc shape on the 3T MR unit, which is superior in detecting subtle anatomical differences to lower fields. The most common disc shape in our study was biconcave, concordantly to other studies [31,32]. The second most common shape was biplanar, followed by hemiconvex, biconvex, and, most rarely, deformed. A biconcave disc is most often found in normal findings on an MRI and more often found in anterior DDWR compared with DDWOR. In other words, a normal, biconcave disc shape is rarely found in advanced cases of TMD. A biconvex disc (as well as deformed), on the other hand, was mostly found in anterior DDWOR, speaking to a severe ongoing degenerative process. These results are in accordance with other studies [33,34]. Disc dislocation gradually leads to the shortening of the disc and the loss of biconcave shape, and it eventually becomes an obstacle for jaw opening. This explains the finding of biconvex and deformed short discs in more advanced cases of TMD. In the posterior disc dislocation, the shape of the disc was most often biplanar, significantly different compared with other disc shapes. This is also in concordance with previous studies [35]. A biplanar disc does not have the eminence that disables posterior dislocation, so this shape represents a risk factor for this type of dislocation.

The mean angle of the articular eminence ranges from 30 to 60°, with the FH used as a reference plane [36]. In our study, the mean angle of the articular eminence was 47°, independent of the presence of disc dislocation (slightly, insignificantly greater in anterior DDWOR, 48°). The articular eminence is reported to be steeper in anterior dislocations [37]. The steeper articular eminence (as a consequence of acquired conditions such as tooth loss) was considered to be a risk factor for TMD development [38]. Our study failed to show a significant correlation between the eminence inclination and the presence or type of disc dislocation, similar to the study by Shahidi [37]. As well as in previous studies, no significant difference between various depths of GF and types of disc dislocation was observed [39]. The analysis of the dimensions of anatomical structures in absolute values is susceptible to bias since it is highly dependent on the personal habitus. Unfortunately, to date, there is no adequate numerical model for relative measurement [31].

The agreement between the diagnosis established on MR examination and the clinical finding is variable, ranging from 59 to 90% [40,41]. Some authors doubt the possibility to accurately assess the disc position on the clinical examination [42]. On the other hand, the number of false positive diagnoses on MRI is high [41]. There is a lack of published data on the specificity and sensitivity of the RDC/TMD protocol compared with MRI examination, especially on 3T field strength. In the study of Manoliu et al., it was concluded that high-resolution 3T magnetic field strengths have better overall image quality, better visibility, and delineation of both the TMJ disc and the part near to the condylar neck of the pterygoid muscle [43]. Our study confirmed a high specificity of the RCD/TMD diagnostic criteria but a low sensitivity, compared with the gold standard (MRI) (Table 3). In other words, there is a small chance to establish a false positive diagnosis of anterior DDWR and DDWOR using RDC/TMD only. The clinical examination of the TMJ revealed disc dislocation in 9.2% of the subjects, while the MRI confirmed signs of disc dislocation in 17.3%. On clinical examination, only 3% of the patients had signs of bilateral disc dislocation, while on the MRI, a bilateral condition was detected in 16% of the patients. The low sensitivity of the clinical examination is the consequence of the high number of false negative diagnoses. In a number of cases, the clinical finding according to the RDC/TMD criteria was normal, while disc dislocation was observed on the MRI. Our results are concordant to the studies by Marpaung and others [44]. The MRI is superior in establishing the diagnosis of disc dislocation in asymptomatic patients (in our study, 18% of the patients were confirmed to have disc dislocation based on the MRI findings) since it is capable of presenting pathological changes long before they clinically manifest. According to the literature, this percent can be even higher, up to 40% [45]. This raises the question whether in some patients, disc dislocation can be anatomical variety, rather than a pathological condition [46]. Some authors reported good agreement between radiological and clinical findings [40], while others, such as our study, presented poor agreement [39]. In studies with a low level of agreement, the majority of the patients were asymptomatic, concordantly with our study, where fair agreement was observed also. However, the kappa coefficient was somewhat higher in anterior DDWOR. It could be explained by the fact that this dislocation is more commonly found in more severe cases of TMD that are usually both symptomatic and clinically detectable. Although it may seem that the RDC/TMD protocol is not accurate enough in detecting disc dislocations, it is necessary to highlight that the majority of disc dislocations found on the clinical exam were confirmed on the subsequent MR examination. The participants in our study were asymptomatic, and 11.2% were diagnosed with disc dislocation based on the findings on clinical examination. DDWR is a common disorder in TMJ, but it is most often a stable, painless, and lifelong condition, leading to the loss of reduction in a very small proportion of patients. For such a condition, it is more important for the clinical tool to be specific than sensitive, in order to avoid a false positive diagnosis and unnecessary treatment [46]. There are some efforts recently dedicated to the new scoring system, based on the MRI findings as the gold standard. This novel staging system is able to demonstrate the state of the LPM, joint effusion, degenerative disc changes, and disk perforation, as well as bone degeneration and translation of the condyle, and also the integrity of the retrodiscal layers in addition to the direction of disc displacement [47]. The authors of this system confirmed that patients can be assigned into the right stage of TMD based on these parameters.

The agreement between the clinical and radiological findings for osteoarthritis was even lower (with very low sensitivity), owing partly to the fact that MRI is not a method of choice for depicting osseous changes [48]. More importantly, MRI demonstrates the degenerative changes years before they clinically manifest [49]. Finally, osteoarthritic changes are followed by transient pain, present only in the active phase of the disease [50].

### Limitations

This study has some limitations. The described random recruitment of the participants, independent of clinical signs and symptoms, could lead to a certain level of bias. The influence of comorbidities, such as chronic pain or depression, needs to be considered and remains as a recommendation for future studies. The second limitation was that we included only female participants, so conclusions could not be observed as being population based. Considering the design of the MRI protocol, it should be mentioned that the images were not acquired in the coronal position of TMJ on the MRI, so the potential lateral component of dislocation was not evaluated. However, the lateral dislocation is mainly the result of the traumatic injury [51], and the subjects in our study were asymptomatic and randomly selected, so the presence of lateral dislocation was not an expected finding.

The most recent advances in MRI protocols could also help in improving the reliability of imaging in confirming clinical findings, especially the dynamic MRI sequences, which are suggested to be more beneficial for the evaluation of the morphology and function of the TMJ compared with static sequences, especially in patients with TMJ disc displacement [20,52].

## 5. Conclusions

The agreement between the clinical findings of pathological processes and the detection on MRI is fair, especially in the means of anterior disc dislocation with reduction, where the sensitivity of clinical examination remains low compared with 3T MRI. However, the number of false positive diagnoses established using RDC/TMD is low, especially in the group of asymptomatic patients, thus limiting unnecessary treatment. Nevertheless, in patients who report symptoms that could be explained by changes in the TMJ for which clinical examination shows low sensitivity, it would be reasonable to include an MRI examination at a certain point of the management and diagnostics. The RDC/TMD criteria remain a sensible method for both establishing a relevant clinical diagnosis and avoiding the overtreatment of patients. However, further improvements in the field of diagnostic criteria are desirable in order to increase the sensitivity of this method. The inclusion of multidisciplinary intervention for the management of TMD pain that would take into account all the neuromuscular structures of the masticatory system, findings on the clinical examination, self-reported symptoms, and findings on the MRI might be the desirable direction of future research. In addition, constant training and calibration of the clinical examiners will certainly improve the diagnostic capability, sensitivity, and specificity of the clinical findings.

## Figures and Tables

**Figure 1 diagnostics-13-01986-f001:**
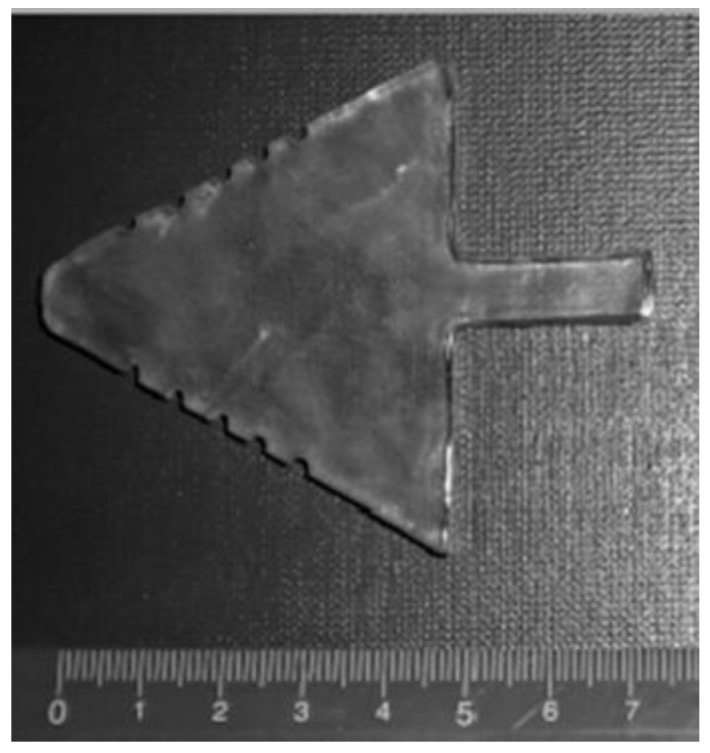
Mechanic arrow-like mouth opener, with incisures on upper and lower surfaces (places to put incisors inside) at distances of 15 mm, 20 mm, 25 mm, 30 mm, and 35 mm.

**Figure 2 diagnostics-13-01986-f002:**
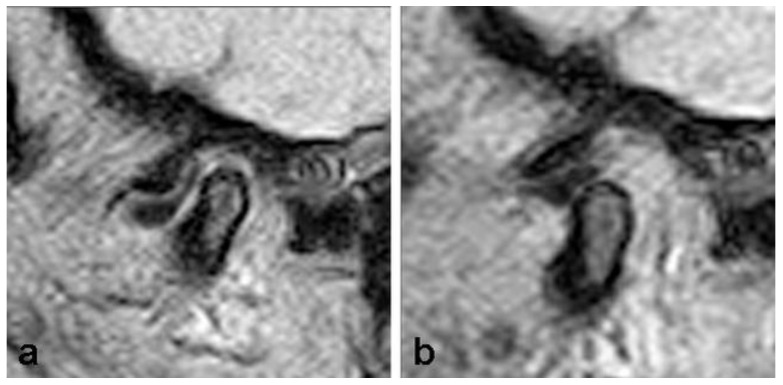
Anterior disc dislocation with reduction with closed (**a**) and opened (**b**) jaws. With the closed jaw, the posterior zone of the disc is located in front of the 12 o’clock position, while with the opened jaw, it takes a normal position.

**Figure 3 diagnostics-13-01986-f003:**
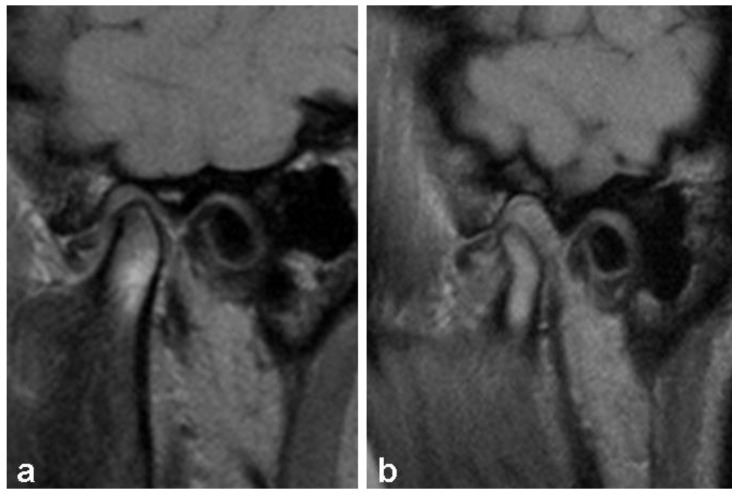
Anterior disc dislocation without reduction with closed (**a**) and opened (**b**) jaws.

**Figure 4 diagnostics-13-01986-f004:**
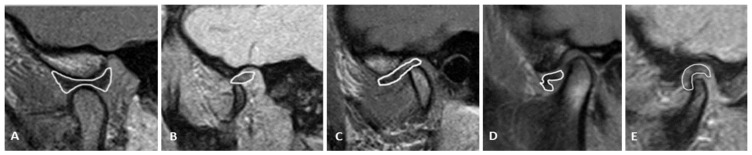
Disc shape, based on the Murakami classification: (**A**) biconcave–normal disc, (**B**) biconvex, (**C**) biplanar, (**D**) hemiconvex, and (**E**) folded or deformed.

**Figure 5 diagnostics-13-01986-f005:**
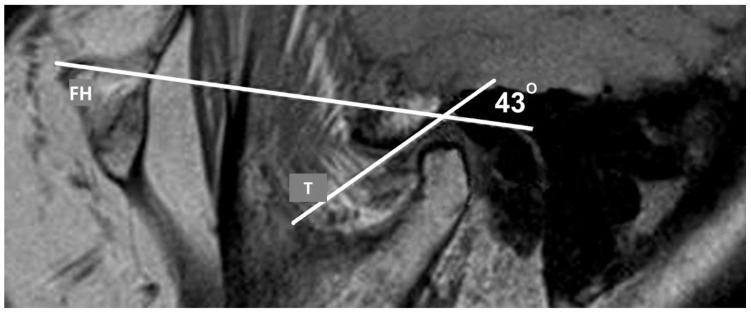
Parasagittal PD MR image of the TMJ with measurement of the articular eminence angle in the position of the closed jaw. FH—Frankfort horizontal plane; T—tangent, angle 43°.

**Figure 6 diagnostics-13-01986-f006:**
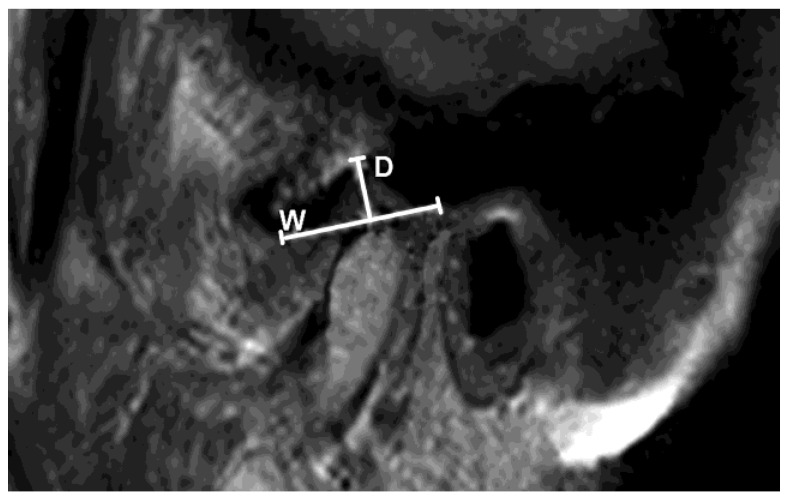
Parasagittal PD MR image of the TMJ represents measurement of glenoid fossa depth (D): value between highest point of glenoid fossa and line representing fossa width (W).

**Figure 7 diagnostics-13-01986-f007:**
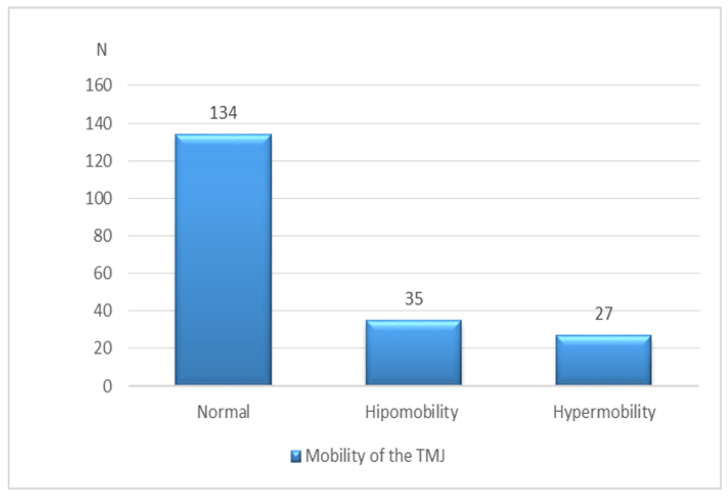
Mobility of TMJ.

**Table 1 diagnostics-13-01986-t001:** Clinical and MRI findings in participants.

Diagnosis based on RDC/TMD ^1^ Criteria	N (%)	Diagnosis based on MR Imaging	N (%)
Normal	170 (86.7)	Normal	99 (50.5)
Disc dislocation with reduction	13 (6.6)	Anterior dislocation with reduction	23(11.7)
Disc dislocation without reduction	5 (2.6)	Anterior dislocation without reduction	9 (4.6)
		Posterior dislocation	2 (1,0)
Arthralgia/Osteoarthritis/Osteoarthrosis	5 (2.6)	Osteoarthritis	46 (23.5)
Arthralgia/Osteoarthritis/Osteoarthrosis+ Disc dislocation with reduction	2 (1.0)	Anterior dislocation with reduction + Osteoarthritis	10 (5.1)
Arthralgia/Osteoarthritis/Osteoarthrosis+ Disc dislocation without reduction	1 (0.5)	Anterior dislocation without reduction + Osteoarthritis	10 (5.1)
		Posterior dislocation + Osteoarthritis	4 (2.0)
Total	196 (100)	Total	196 (100)

**^1^** Research Diagnostic criteria for temporomandibular disorders.

**Table 2 diagnostics-13-01986-t002:** Data on depth of GF, angle of articular eminence, and disc length with regard to disc location.

		N	Mean	SD	Minimum	Maximum	F	*p*
**Depth of GF (mm)**	Normal	145	4.00	1.10	2.00	7.00	1.417	0.245
	Anterior dislocation with reduction	33	3.88	1.11	2.00	6.00		
	Anterior dislocation without reduction	12	4.50	1.17	3.00	6.00		
	Total	190	4.01	1.11	2.00	7.00		
**Angle of articular eminence (^o^)**	Normal	145	46.36	6.37	31.00	64.00	0.531	0.589
	Anterior dislocation with reduction	33	47.18	6.68	34.00	61.00		
	Anterior dislocation without reduction	12	48.00	5.39	39.00	55.00		
	Total	190	46.61	6.36	31.00	64.00		
**Disc length** **(mm)**	Normal	145	9.98	1.73	6.00	14.00	1.188	0.307
	Anterior dislocation with reduction	33	9.46	1.58	6.00	12.00		
	Anterior dislocation without reduction	12	10.00	2.76	4.00	13.00		
	Total	190	9.89	1.79	4.00	14.00		

**Table 3 diagnostics-13-01986-t003:** Agreement between clinical and magnetic resonance imaging findings on anterior disc dislocation with and without reduction.

	Sensitivity	Specificity	K	*p*
Normal				
Anterior DDWR	24.2%	94.9%	0.240	<0.001
Anterior DDWOR	25.0%	99.4%	0.355	<0.001
Disc dislocation overall	31.1%.	95.9%	0.301	<0.001
Osteoarthritis of the TMJ	4.8%	96.2%	0.013	0.740

## Data Availability

The data presented in this study are available in this article.

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
