# Peer review of "High-Field Magnetic Resonance Imaging of the Temporomandibular Joint Low Agreement with Clinical Diagnosis in Asymptomatic Females"

_diagnostics, 2023, doi:10.3390/diagnostics13121986_

Round 1

Reviewer 1 Report

There is a lack of published data on specificity and sensitivity of RDC/TMD protocol compared to MRI examination, especially on 3T field  strength. Therefore, this paper deserves publication.

Author Response

- Dear reviewer #1,

Thank you for your kind remark and for revising our submitted manuscript.

Reviewer 2 Report

In the introduction section, the authors should also discuss TMJ dysfunctions secondary to head/spine vicious posture, also further complications. The biomechanics of TMJ should also be approached.

The authors should provide the sample size calculation power and effect size.

The authors could also provide detailed data regarding age or gender differences in their sample sizes. Differences between clinical and MRI diagnosis should be mentioned by providing the effect size (Cohen”s D value).

The authors must specify if they checked the normality of data distribution.

Lines 232-246 should be moved into the Introduction section.

Most of the references are outdated, therefore, they must be updated.

Minor English editing.

Author Response

Reviewer #2:

  1. In the introduction section, the authors should also discuss TMJ dysfunctions secondary to head/spine vicious posture, also further complications. The biomechanics of TMJ should also be approached.

- Dear Reviewer #2,

The authors agree with this point. We added the discussion of TMD secondary to the changes in head/spine posture as well as a short insight into TMJ biomechanics in the introduction section of the Manuscript.

  1. The authors should provide the sample size calculation power and effect size.

-Sample size was calculated by GPower (G*Power, Version 3.1.9.6, Heinrich Heine

Universität, Düsseldorf, Germany). The number of participants needed to achieve a

statistical power of .80 for medium-size effect and an α error probability of 0.05 was 145.

  1. The authors could also provide detailed data regarding age or gender differences in their sample sizes. Differences between clinical and MRI diagnosis should be mentioned by providing the effect size (Cohen”s D value).

-Since the aim of the study did not involve testing differences between groups, the authors had

no reason to test differences in age. The research sample included only women. In the

research, we did not use tests of differences between groups, so we did not calculate

accordingly Cohen´s d (Cohen´s d is an effect size used to indicate the standardised difference

between two means).

  1. The authors must specify if they checked the normality of data distribution.

- Thank You for this observation. Distribution of data was normal for all variables.

Angle of articular eminence skewness= -0,523, kurtosis=-0,343; Disc lenght skewness= -0,088,   kurtosis=0,153; Depth of glenoid fossa skewness=0,082, kurtosis= -0,729.

  1. Lines 232-246 should be moved into the Introduction section.

-Thank You, we moved Lines 232-246 in the introduction section.

  1. Most of the references are outdated, therefore, they must be updated.

-Thank You for this observation, we further updated reference list.

Reviewer 3 Report

The given abstract presents the key aspects of a study that aims to investigate the agreement between clinical diagnosis based on Research Diagnostic Criteria/Temporomandibular disorders (RDC/TMD) and high-field magnetic resonance imaging (MRI) findings in temporomandibular joints (TMJ) of asymptomatic women. The abstract provides information on the study's objectives, methods, results, and conclusions.

The methods section outlines the study design, sample size, and the utilization of clinical examination according to RDC/TMD criteria, alongside same-day MRI using a 3T MR unit. However, the abstract lacks specific details regarding the inclusion criteria for the asymptomatic women, potentially affecting the generalizability of the results.

The results section presents the prevalence of TMJ pathologies as determined by clinical criteria and MRI findings. The abstract highlights that anterior disc dislocation with reduction showed lower agreement between clinical and MRI findings compared to disc dislocation without reduction, indicating limitations in clinical examination sensitivity. Sensitivity values for various conditions, including osteoarthritic changes, are reported as low, while specificity remains high. These findings contribute to understanding the strengths and weaknesses of clinical diagnosis and MRI in detecting TMJ pathologies.

The abstract's conclusion states that clinical examination sensitivity remains low compared to 3T MRI, especially for osteoarthritic changes and anterior disc dislocation with reduction. It emphasizes the low number of false positive diagnoses using RDC/TMD, suggesting that it is a reasonable method for clinical diagnosis and the prevention of overtreatment in asymptomatic patients.

The abstract effectively summarizes the main findings of the study, highlighting the importance of utilizing MRI for enhanced sensitivity in diagnosing TMJ pathologies. However, it lacks specific details regarding the statistical analyses performed, such as the descriptive statistics and post hoc analysis mentioned in the methods section. Including these details would provide a better understanding of the study's methodology and enhance the abstract's comprehensibility.

Additionally, the abstract would benefit from addressing the implications of the study's findings for clinical practice. It could discuss how the results could potentially influence treatment decisions or the need for further research to improve diagnostic criteria.

In summary, while the abstract provides a clear overview of the study, it could be further improved by including additional methodological details, discussing the implications of the findings, and refining the language for better readability. Considering these suggestions, I would rate the given abstract as 6.5 out of 10.

Author Response

Reviewer #3:

  1. The given abstract presents the key aspects of a study that aims to investigate the agreement between clinical diagnosis based on Research Diagnostic Criteria/Temporomandibular disorders (RDC/TMD) and high-field magnetic resonance imaging (MRI) findings in temporomandibular joints (TMJ) of asymptomatic women. The abstract provides information on the study's objectives, methods, results, and conclusions.

- Thank you for your kind comments and revision of our submitted manuscript.

  1. The methods section outlines the study design, sample size, and the utilization of clinical examination according to RDC/TMD criteria, alongside same-day MRI using a 3T MR unit. However, the abstract lacks specific details regarding the inclusion criteria for the asymptomatic women, potentially affecting the generalizability of the results.

- Thank you for this observation, we added inclusion criteria into the methods section of the abstract.

  1. The results section presents the prevalence of TMJ pathologies as determined by clinical criteria and MRI findings. The abstract highlights that anterior disc dislocation with reduction showed lower agreement between clinical and MRI findings compared to disc dislocation without reduction, indicating limitations in clinical examination sensitivity. Sensitivity values for various conditions, including osteoarthritic changes, are reported as low, while specificity remains high. These findings contribute to understanding the strengths and weaknesses of clinical diagnosis and MRI in detecting TMJ pathologies.

- Thank you for this observation.

  1. The abstract's conclusion states that clinical examination sensitivity remains low compared to 3T MRI, especially for osteoarthritic changes and anterior disc dislocation with reduction. It emphasizes the low number of false positive diagnoses using RDC/TMD, suggesting that it is a reasonable method for clinical diagnosis and the prevention of overtreatment in asymptomatic patients.

- Thank you for this observation.

  1. The abstract effectively summarizes the main findings of the study, highlighting the importance of utilizing MRI for enhanced sensitivity in diagnosing TMJ pathologies. However, it lacks specific details regarding the statistical analyses performed, such as the descriptive statistics and post hoc analysis mentioned in the methods section. Including these details would provide a better understanding of the study's methodology and enhance the abstract's comprehensibility.

- After careful evaluation of the text, we added the details on statistical methods that, in our opinion, will improve the comprehensiveness of the manuscript.

  1. Additionally, the abstract would benefit from addressing the implications of the study's findings for clinical practice. It could discuss how the results could potentially influence treatment decisions or the need for further research to improve diagnostic criteria.

- The authors completely agree with the Reviewer. However, due to word count restrictions, we decided to add the discussion on potential influence on clinical management of the patient as well as recommendations for the future studies (or extension of our own study) in the Conclusion part of the manuscript.

  1. In summary, while the abstract provides a clear overview of the study, it could be further improved by including additional methodological details, discussing the implications of the findings, and refining the language for better readability. Considering these suggestions, I would rate the given abstract as 6.5 out of 10.

- The authors thank the reviewer for honest scoring of the manuscript quality. We did our best to refine the language in order to improve readability and to add or repeat critical information in order to make the manuscript more straightforward.

Round 2

Reviewer 2 Report

The authors performed the requested adjustments, therefore I consider the paper suitable for publication.